# Brief Communication: Lower Bound Estimates for Residence Time of Energy in the Atmospheres of Venus, Mars and Titan

Javier Pelegrina[1], Carlos Osácar[1], and Amalio Fernández-Pacheco[2]

[1]Facultad de Ciencias. Universidad de Zaragoza. 50009 Zaragoza (Spain)
[2]Facultad de Ciencias and BIFI. Universidad de Zaragoza. 50009 Zaragoza (Spain)

**Correspondence:** Carlos Osácar (cosacar@unizar.es)

**Abstract.**

The residence time of energy in a planetary atmosphere, $\tau$, recently introduced and computed for the Earth's atmosphere (Osácar et al., 2020), is here extended to the atmospheres of Venus, Mars and Titan. $\tau$ is the timescale for the energy transport across the atmosphere. In the cases of Venus, Mars and Titan, these computations are mere lower bounds due to a lack of some energy data. If the analogy between $\tau$ and the solar Kelvin-Helmholtz scale is assumed, then $\tau$ would also be the time the atmosphere needs to return to equilibrium after a global thermal perturbation.

## 1 Introduction

When the inflow, $\mathcal{F}_i$, of any substance into a box is equal to the outflow, $\mathcal{F}_o$, then the amount of that substance in the box, $\mathcal{M}$, is constant. This constitutes an equilibrium or steady state. Then, the ratio of the stock in the box to the flow rate (in or out) is called residence time and is a timescale for the transport of the substance in the box.

$$t = \frac{\mathcal{M}}{\mathcal{F}}. \tag{1}$$

In equation (1) it is assumed that the substance is conserved. A good example of this type is the parameter defined in atmospheric chemistry (Hobbs, 2000) as the average residence time of each individual gas, defined as (Eq. 1). $\mathcal{M}$ is the total average mass of the gas in the atmosphere and $\mathcal{F}$ the total average influx or outflux, which in time average for the whole atmosphere are equal.

In this work we extend the substance that flows in the box from matter to energy and the residence time is

$$\tau = \frac{E}{F}, \tag{2}$$

where $E$ is the total energy in the box (a planetary atmosphere) and $F$ the energy flux that enters or leaves it.

Here, by using (2), we estimate the average residence time of energy in several planetary atmospheres. Planetary atmospheres constitute steady state problems because the storage of energy in their interior is not systematically increasing or decreasing. For completeness, it is worth recalling that several authors have previously considered the energy-residence time relation in other type of problems (Mcilveen, 1992, 2010; Harte, 1988).

The structure of this communication is as follows: Section 2 addresses the numerator of Eq. (2) $E$, while Section 3 deals with the denominator $F$. In Section 4 the residence time of energy is considered for the Sun. In Section 5, the radiative constant is introduced and compared with the atmospheric residence timescale. In Section 6 we comment on some final points.

## 2  Forms of energy in a planetary atmosphere

The most important forms of energy in an atmosphere are: the thermodynamic internal energy, $U$, the potential energy due to the planet's gravity, $P$, the kinetic energy, $K$, and the latent energy, $L$, related to the phase transitions.

In a planar atmosphere, in hydrostatic equilibrium and by using the state equation for an ideal gas, these magnitudes can be written as

$$U = \int_0^\infty c_v\, T(z)\, \rho(z)\, \mathrm{d}z = \frac{c_v}{R} \int_0^\infty p(z)\, \mathrm{d}z, \tag{3}$$

$$P = \int_0^\infty g\, z\, \rho(z)\, \mathrm{d}z = \int_0^\infty p(z)\, \mathrm{d}z, \tag{4}$$

In expressions (3) and (4), $c_v$ is the specific heat at constant volume, $R$ is the gas constant and $\rho(z)$ and $T(z)$ are the density and temperature of the mixture of gases of the atmosphere, respectively. E stands for the total energy in the atmosphere:

$$E = U + P + K + L. \tag{5}$$

The sum $S = U + P$ will be called dry static energy, then

$$E = S + K + L. \tag{6}$$

It is important to remark that $S$ is much bigger than the sum $K + L$. For example, for the Earth (Peixoto and Oort, 1992)

$$\frac{S}{K + L} = \frac{150}{6} = 25. \tag{7}$$

In the case of Earth's atmosphere, the four terms $U$, $P$, $K$, and $L$ are known (Peixoto and Oort, 1992), so we know $E$. However for the atmospheres of Venus, Mars and Titan we can only compute the terms $U$ and $P$ and estimate $S$ but not $E$. We have carried out these computations by performing the numerical integration (4), using the vertical data $p(z)$ shown in (Sánchez-Lavega, 2011, page 212-227). The results of $E$ or $S$ for each planet are shown in Table 1.

For the Earth's atmosphere, the estimates of different authors are very similar. Table 2 compares values of Peixoto and Oort (1992) and Hartmann (1994). The last row corresponds to the difference between the total energy of the Earth's atmosphere ($E$) and its dry static energy ($S$). The kinetic and latent components can be neglected in a first approximation.

The sound velocity of an ideal gas is

$$c = \sqrt{\gamma \frac{R*}{M} T} \tag{8}$$

**Table 1.** Forms of energy in planetary atmospheres

|  | Venus | Earth | Mars | Titan |
|---|---|---|---|---|
| $P\,(\mathrm{J\,m^{-2}})$ | 1.24E+11 | 7.00E+08 | 6.05E+06 | 2.63E+09 |
| $U\,(\mathrm{J\,m^{-2}})$ | 4.31E+11 | 1.80E+09 | 2.10E+07 | 6.79E+09 |
| $S\,(\mathrm{J\,m^{-2}})$ | 5.55E+11 | 2.50E+09 | 2.71E+07 | 9.42E+09 |
| $K\,(\mathrm{J\,m^{-2}})$ | ... | 1.30E+06 | ... | ... |
| $L\,(\mathrm{J\,m^{-2}})$ | ... | 7.00E+07 | ... | ... |
| $E\,(\mathrm{J\,m^{-2}})$ | ... | 2.57E+09 | ... | ... |
| $C_p/R$ | 4.47 | 3.5 | 4.37 | 3.58 |

**Table 2.** Earth's energy comparison

| Units $10^6\,\mathrm{J\,m^{-2}}$ | Peixoto and Oort (1992) | Hartmann (1994) | $\Delta(\%)$ |
|---|---|---|---|
| P | 693 | 700 | 0.17 |
| U | 1803 | 1800 | -1.01 |
| L | 63.8 | 70 | -9.72 |
| K | 1.23 | 1.3 | -5.69 |
| E | 2561 | 2571 | -0.39 |
| S | 2493 | 2500 | -0.28 |
| $(E-S)/E\,(\%)$ | 2.539 | 2.773 |  |

where $R^*$ is the universal constant of gases and $M$ is the molecular mass of the gas; $\gamma = C_p/C_v$ is the adiabatic constant and $T$ the temperature. The sound velocity can be used to estimate the ratio between $K$ and $S$.

$$\frac{K}{S} \approx \left(\frac{v}{c}\right)^2 \tag{9}$$

In the case of Mars, on surface $c = 228.73\,\mathrm{m\,s^{-1}}$. Table 3 contains data of winds measured by Viking probes on the surface (Sheehan, 1996, p. 194). With these data, $K$ can be neglected in Mars. In the case of Titan, Mitchell (2011) assumes that the kinetic energy can be neglected. Based on these figures, the kinetic energy can be omitted in a first approximation for Mars and Titan.

In case when $S$ is not much bigger than $K+L$, our results for $\tau$ would be a lower bound. Future observations will determine these numbers.

**Table 3.** Wind velocity in Mars

|  | Day | Night | Storm | Max during storm |
|---|---|---|---|---|
| $v$ (m/s) | 7 | 2 | 17 | 26 |
| $K/S \approx (v/c)^2$ | 0.0009 | 0.00007 | 0.0055 | 0.0129 |

**Table 4.** Fluxes of energy and residence times in planetary atmospheres

|  | Venus | Earth | Mars | Titan |
|---|---|---|---|---|
| $F_i\,(\mathrm{W\,m^{-2}})$ | $17292 \pm 1715$ | $561 \pm 9.17$ | $49 \pm 3.97$ | $6.88$ |
| $F_o\,(\mathrm{W\,m^{-2}})$ | $17292 \pm 1713$ | $561 \pm 5$ | $49 \pm 4.239$ | $6.87$ |
| $\tau\,(\mathrm{days})$ | $371.48 \pm 26.04$ | $53.43 \pm 0.42$ | $6.87 \pm 0.41$ | $15916$ |

## 3 Energy fluxes absorbed and emitted by the planetary atmospheres and residence times

The values of the energy fluxes for all planets have been deduced from Read et al. (2016). For each planet, $F_i$ and $F_o$ represent the inflow and outflow of energy absorbed or emitted by the atmospheres. The so called 'Trenberth diagrams' are particularly suited to the identification of these fluxes.

As an example, in the case of Venus (see Read et al. (2016, Figure 6)), the fluxes absorbed by the atmosphere ($F_i$) are: $135\,\mathrm{W\,m^{-2}}$ from incoming solar radiation (shortwave) absorbed in the middle atmosphere, $3\,\mathrm{W\,m^{-2}}$ from incoming solar

radiation absorbed by the lower atmosphere; and $17154\,\mathrm{W\,m^{-2}}$ of longwave flux absorbed from surface. Thus, the total influx is $17292\,\mathrm{W\,m^{-2}}$.

The emitted fluxes ($F_o$) are $17132\,\mathrm{W\,m^{-2}}$ of longwave radiation to surface and $160\,\mathrm{W\,m^{-2}}$, of longwave radiation emitted from atmosphere to space. The total outflux value is $17292\,\mathrm{W\,m^{-2}}$. Analogous calculations for the rest of planets give the values for $F_i$ and $F_o$ shown in Table 4.

These energy fluxes were computed by Read et al. (2016) through complex and detailed numerical models. Their results coincide well with observations and have little uncertainty, so its effect on the residence time of energy can be neglected. In any case, here we have computed that uncertainty value.

For Earth, quoting (Read et al., 2016, p. 704): *"the Earth's energy budget has been quantified in the most detail and to relatively high precision (...). Even so, a number of significant uncertainties persist (...). The incoming solar flux (or solar*

*irradiance) is known to the highest accuracy at $340.2\,Wm^{-2}$ (Kopp and Lean, 2011) and varies the least of all the other fluxes. For the other fluxes, estimates vary as to their likely uncertainty, from around $1\,Wm^{-2}$ for some to around $10\,Wm^{-2}$ for the least well-characterized quantities (...). Figure 1 summarizes the recent set of estimates obtained from combinations of remote sensing and in situ measurements, together with well-validated numerical model simulations (...). These represent some of the most comprehensive studies to date that include strenuous efforts to trace the uncertainties in all of the main fluxes. (...) Figure*

*1 thus represents the current state of the art in deriving such an energy budget for an entire planet."* Although (Read et al., 2016) do not give exact numbers for uncertainty of energy fluxes, their references herein do. We have computed the following uncertainty values: $F_{in} = 561 \pm 9.17\,\mathrm{W\,m^{-2}} \Rightarrow \tau = 53.43 \pm 0.87$ d, and $F_{out} = 561 \pm 5\,\mathrm{W\,m^{-2}} \Rightarrow \tau = 53.43 \pm 0.48$ d. We note how both fluxes and residence times are extremely similar and compatible. A weighted average would give us $\tau = 53.43 \pm 0.42$ d.

When computing the energy fluxes of Mars, Read et al. (2016) use a detailed radiative transfer model *"suggesting an uncertainty in infrared fluxes of around 6–12%"*. By using the worst case scenario of a 12% uncertainty, we obtain $F_{in} =$

$49 \pm 3.97$ Wm$^{-2} \Rightarrow \tau = 6.87 \pm 0.56$ d, and $F_{out} = 49 \pm 4.23$ Wm$^{-2} \Rightarrow \tau = 53.43 \pm 0.59$. This gives us $\tau = 6.87 \pm 0.41$ d. These uncertainties are reflected in Table 4.

About the energy fluxes in Venus, Read et al. (2016) state: *"energy fluxes agree with available observations to around* $\pm 10\%$*"*. However, they admit that "*the energy budget presented (...) should therefore be seen as a plausible scheme that is internally self-consistent and representative of a reasonably good radiative–dynamical model of the Venus atmosphere in equilibrium*". Assuming an uncertainty of 10% in energy fluxes, $F_{in} = 17292 \pm 1715$ Wm$^{-2} \Rightarrow \tau = 371.48 \pm 36.84$ d, and $F_{out} = 17292 \pm 1713$ Wm$^{-2} \Rightarrow \tau = 371.48 \pm 36.80$ d. This gives $\tau = 371.48 \pm 26.04$ d.

In Titan's energy fluxes, Read et al. (2016) do not state any bound on uncertainties. However, they say (Read et al., 2016, p.711) *"energy fluxes are consistent with the measurements of Li et al. (2011) to within a few per cent, although the internal and surface fluxes are not well constrained by observations."*. We can assume that the energy fluxes they present and used here, are fairly accurate with low uncertainty.

With the total energy values, $E$ or $S$ (in Table 1) and $F$ (Table 4), we estimate the value of residence time of energy in the atmosphere of each planet. However, as we stressed above, strictly speaking $E$ is only known in the Earth's case. In the other three cases, the ratio $(S/F)$ is a lower bound for the actual residence time.

$$\frac{S}{F} \leq \frac{E}{F} = \tau. \tag{10}$$

These results and their estimated uncertainties are shown in Table 4.

## 4   Residence time of energy in the Sun

The Sun is also in a steady state for the energy. The temperatures in its interior are not systematically increasing or decreasing. In Stix (2003) it is shown that in solar physics, the Kelvin-Helmholtz timescale (KH) corresponds to both the time that a photon takes from the core until it leaves the surface and the time necessary for the star to return to equilibrium after a global perturbation (Kippenhahn and Weigert, 1994):

$$\tau_{KH} = \frac{GM_\odot^2}{R_\odot L_\odot} \sim 10^7 \, \text{yr}. \tag{11}$$

For more details on the residence time of energy in the Sun see Osácar et al. (2020). Furthermore, Spruit (2000) shows that KH is the longest timescale for any solar perturbations.

In summary, if the analogy between the solar KH and the atmospheric $\tau$ is assumed, then $\tau$ is not only the timescale for the energy transport, but also the timescale the atmosphere needs to return to equilibrium after a global thermal perturbation. Furthermore, $\tau$ is the longest timescale for any atmospheric perturbation.

## 5   Radiative relaxation timescale

The most simple models that can be devised for the structure of the atmosphere are the static radiative ones. But if energy transfer with the surface is taken into account, the structure produced under radiative equilibrium cannot be maintained. Con-

vection develops spontaneously and neutralizes the stratification introduced by radiative transfer. The new radiative–convective equilibrium produces two layers. Below a certain height, the thermal structure is controlled by convective overturning and constitutes the troposphere. In this layer, the vertical profile of temperature is adiabatic. In the layer above troposphere, which constitutes the stratosphere, the thermal structure remains close to radiative equilibrium, because radiative transfer stabilizes the stratification.

In general, if a state of equilibrium is perturbed, the atmosphere uses the most efficient mechanism at hand to neutralize it. The mechanism can be convective, advective or radiative. $\tau_R$ is the time it would take to relax the perturbation by radiating the energy excess in the infrared.

A pertubative computation, see for example Wells (2012), gives

$$\tau_R = \frac{c_p p/g}{4\sigma T_{\text{eff}}^3}. \tag{12}$$

In this expression, $p$ is pressure, $c_p$ is the specific heat at constant pressure, $g$ is gravity, $\sigma$ is the Stefan-Boltzmann constant and $T_{\text{eff}}$ is the blackbody effective temperature of the planet.

Defining $T_r$ and $p_r$ as the temperature and pressure of the level from which most of infrared photons are emitted to the space, and $T_s$ and $p_s$ as the temperature and pressure at surface, if we assume that in the troposphere the temperature profile is given by a dry adiabat, then we have

$$p_r = p_s \left(\frac{T_r}{T_s}\right)^{\frac{c_p}{R}}. \tag{13}$$

Assuming this hypothesis (Pierrehumbert, 2010), for the Earth, we obtain $p_r = 670\,\text{mb}$. The Earth's actual $p_r$ is somewhat lower than this estimate because the tropospheric temperature decays less strongly with height than the dry adiabat. For this value of $p_r$, the value of $\tau_R$ is about 22 days.

Due to the factor $p$ in the numerator of Eq. 12, the value of $\tau_R$ decreases rapidly with height. Therefore, radiation is not an efficient mechanism to neutralize perturbations in the low troposphere. In that region $\tau_R$ is thus very long.

We find a clear example of these phenomena is in Venus, where $\tau_R$ varies from 116 days at 40 Km (lower cloud deck) to 0.5 hr at 100 Km (Sánchez-Lavega et al., 2017).

If the above mentioned analogy between the atmospheric $\tau$ and the solar scale KH is assumed, $\tau$ is the time necessary to return to equilibrium after a global perturbation, whilst $\tau_R$ is the timescale corresponding to small perturbations and also $\tau > \tau_R$.

## 6 Final comments

In our opinion, the concept of "Residence time of energy in a planetary atmosphere" is completely original. This residence time has been computed for the atmospheres of Venus, Earth, Mars and Titan. In the cases of Venus, Mars and Titan, they are mere lower bounds due to a lack of data about kinetic and latent energies.

The analogy between $\tau$ and the KH solar timescale seems likely, although this does not constitute a proof.

The usual radiative timescales $\tau_R$ presented by other authors (e.g. Houghton (2002), Wells (2012), Sánchez-Lavega (2011)) are calculated assuming that a small perturbation is produced in the temperature, i.e. it is a perturbative calculus. Furthermore, it depends on the values of pressure and temperature where it is computed. Since about the 80% of radiative flux leaving an atmosphere comes from the cold top of the highest atmospheric opaque layer, we have estimated $\tau_R$ with $T_{\text{eff}}$ at the height of maximum emission, $p_r$. The obtained radiative timescale is smaller than $\tau$. On the contrary, in the computation of residence time of energy $\tau$ in planetary atmospheres, only global averaged planetary parameters are used.

*Data availability.* The data of the energies used for the estimation of residence time in the Venus, Mars and Titan atmospheres were computed with $p$ and $T$ from Sánchez-Lavega (2011, page 212-227). Those for the Earth's atmosphere were extracted from Peixoto and Oort (1992). The fluxes of energy for all the cases were deduced from Read et al. (2016).

*Author contributions.* Amalio Fernández-Pacheco conceived the idea; Carlos Osácar, Javier Pelegrina and Amalio Fernández-Pacheco wrote the paper.

*Competing interests.* The authors declare no conflict of interests.

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
