# Peer review of "Brief Communication: Lower Bound Estimates for Residence Time of Energy in the Atmospheres of Venus, Mars and Titan"

_Nonlinear Processes in Geophysics, 2020_

## Author Response (AR1)

Reply to comments from Referee #1

*This short communication calculates (for some lower bounds) the residence timescale of an atmosphere for Venus, Earth, Mars, and Titan. The authors use the comparability of the residence timescale and the KH timescale to argue that both have the same meaning, i.e., both express the time needed to return to equilibrium after a thermal perturbation. This article can be considered for publication after addressing some concerns listed below.*

*Major comments*

*1   I am Not sure I am convinced that the residence timescale is equivalent to the time needed to return to equilibrium after a thermal perturbation. This claim is from the comparability between the residence timescale and the KH timescale for the Sun. However, this can be a mere coincidence. By definition, the residence timescale is the time that takes, in this case, energy, from entering the system until it goes out. Although it can be, this is not necessary the time to return to equilibrium, other processes, such as radiative, transport, and turbulence, can act to make the equilibration either faster or slower.*

In the new version of the paper, some of the main modifications are:

1.- From the very beginning it is said that the residence time of energy in a planetary  atmosphere , is the time scale of the transport of energy across the atmosphere.
2.- A new distribution of sections has been made.
3.- In the new section 5 it  is explained what the Radiative Relaxation Timescale is.
4.- It is remarked that only if the analogy between $\tau$ and the solar Kelvin–Helmholtz scale is assumed, then $\tau$ would also be the time the atmosphere needs to return to equilibrium after a global perturbation. And it would be the longest timescale because the perturbation had been global.
5.-The analogy between $\tau$ and KH seems likely and attractive but in no way is a proof.
They have been inspired, in part, by your criticism to the first version of the paper.
We acknowledge your comments and dedication.

*2   Lines 22-35: The transition to the Sun is unclear and not motivated. After reading Osacar et al. 2020, it became clear that it is for claiming the residence time as an equilibration timescale. You should give a motivating sentence before this discussion and emphasise the point you want to say.*

A new Section on the residence time of energy in the Sun has been added for clarifying the analogy we assume between KH timescale and $\tau$. See lines 4 and 5 of the first block of the answer. Stix (2003) made the equivalence between residence time and global recovery timescale.

*3   Line 71: "The longest of these scales corresponds to the residence time as computed in Section 3." - Is this statement correct? Can you show or reference this?*

In solar physics -Spruit (2000)- it is shown that KH is the longest timescale. By analogy, in planetary atmospheres, $\tau$ will also be the longest timescale.

> 4 *The radiative timescale discussion is important but feels detached from the rest of the manuscript. You should give a connecting sentence at the beginning of the discussion and a summarising sentence at the end.*

A new Section about the radiative timescale discussion has been added.

> 5 *The parameters you chose are motivated; however, they are not unique. First, the radiative timescale is manifested in our daily living. Consider Earth's seasonal cycle, there is a time-lag of about a month between solstice and the warmest day of the year, resulting from the radiative timescale. Also, Mitchell and Lora (2016) estimate 200 Earth years for the tropospheric radiative timescale of Titan. This radiative timescale is about five times longer than the residence timescale you have calculated. Also, note that there they consider the effect of the atmospheric opacity.*

This time lag between the solstice and the warmest day of the year in a given city depends on the distance between the city and the sea. For this reason, in our opinion, the ocean should be a partner of the model and $\tau_R$ would not be the unique parameter.

The calculation by Mitchell and Lora is done for the Titan's troposphere. By using the formula of Wells (2012) with $p_s=37.9$ mb, $g=1.35$ m s$^{-2}$, $T_e=80$ K, we obtain a radiative timescale of 0.8 years. This computation of $\tau_R$ has been carried out at the pressure where temperature is equal to the blackbody temperature of the planet ($T_{eff}$), where most of the longwave radiation is emitted to space.

Minor comments

> 1 *Line 21: "Harte (1988) uses this concept to estimate the anomalous temperature in urban heat islands." - Adding this line is a bit confusing. Consider elaborating on this point or remove that line.*

We had cited Harte at the beginning of the paper just to show skepticals that the residence time for energy was a valid concept. But, in the same paragraph, we had remarked that the type of problems addressed in this paper was different to that of the temperature of the anomalous heat islands. Though, in a private mail, Prof. J. Harte of Berkeley, expressed his positive opinion about our first paper, we have followed your suggestion and removed that line.

> 2 *Line 48: "It is important to remark that S is much bigger than the sum K + L. For example, for the Earth" - This is probably true, but on other planets, like Mars (or Pluto and Triton) where the atmosphere is thin, or Titan, where it is cold, but still has strong winds, these terms might be comparable. So maybe not be so decisive.*

Yes, if S is not much bigger than K+L, our result would be just a poor lower bound.
Let us remember that K/S ≈ (wind speed / sound speed)$^2$. By using the highest wind speed recorded in Titan of 120 m/s at an altitude of 120 km (see for example Bird, M., Allison, M., Asmar, S. et al. The vertical profile of winds on Titan. Nature 438, 800–802 (2005) https://doi.org/10.1038/nature04060), one obtains that K/S ≈ 0.2.
As wind speeds in Titan decrease steadily as altitude decreases (see Bird et al. (2005)) we are confident that K/S is very small everywhere.

3   *Line 72: "The residence time of energy in a planetary atmosphere characterises the planet, and is computed in a model independent way." - This sentence is unclear. What do you mean by characterises the planet? In what form?*

The usual radiative timescales $\tau_R$ presented by several authors (e.g. Houghton (2002), Wells (2012), Sanchez-Lavega (2011)) are calculated assuming that a small perturbation is produced in the temperature, i.e. it is a perturbative calculus. Furthermore, $\tau_R$ depends on the values of pressure and temperature where it is computed.

On the contrary, in the computation of the residence time of energy $\tau = E/F$ in planetary atmospheres, the unique dependence is on global planetary parameters. It is a property of the whole atmosphere because it is calculated from global averaged parameters.

**6 Reply to comments from Referee #2**

*This short paper presents some simple estimates for what is described as the residence timescale for energy in a planetary atmosphere, as applied to the atmospheres of Venus, Mars and Titan. The calculations are relatively crude, "back of the envelope" estimates based on data derived from the published literature with insufficient detailed explanation or discussion/critique of how accurate or appropriate these are for the purpose described.*

*The motivation for the calculations is also not well developed and the authors seem unaware of the considerable literature on energy storage and transfer in planetary atmospheres.*

*Although this is mentioned in Section 4, how does the proposed timescale differ from the well known radiative relaxation timescale in atmospheric physics (e.g. see J. T. Houghton "The Physics of Atmospheres" Chapter 2 - which is similar to the timescale in Wells 2012)? Such timescales have been computed for many years for all three planets in question as well as for the Earth - e.g. see P Gierasch & R Goody, A study of the thermal and dynamical structure of the Martian lower atmosphere, Plan. Space Sci., 16, 615-646 (1968) for Mars; Pollack JB, Young RE (1975) Calculations of the radiative and dynamical state of the Venus atmosphere. J Atmos Sci 32:1025–1037 for Venus; F. M. Flasar, R. E. Samuelson & B. J. Conrath Titan's atmosphere: temperature and dynamics, Nature, 292, 693-698 (1981) for Titan. For Earth's climate, energetic adjustment timescales have been computed using more sophisticated models - e.g. see T. W. Cronin & K. A. Emanuel, The climate time scale in the approach to radiative-convective equilibrium, JAMES, 5, 843-849 (2013), which takes into account the adjustment timescale for the surface as well as the atmosphere - which seems more appropriate when comparing with the Kelvin-Helmholtz timescale for the Sun. These may not be computing quite the same quantities as what the authors have in mind here, but why not compare them quantitatively with the residence timescale computed here?*

The concept of residence time of energy in a planetary atmosphere, $\tau$, is simple; it is the timescale for the energy transport, and is computed using published data of energy and energy fluxes. Logically its accuracy depends on that of the current experimental data used in the computation.

From a likely comparison with solar physics, we say that if $\tau$ is a timescale similar to the solar KH, we conclude that, after a global thermal perturbation, $\tau$ is also the time that an atmosphere would take to come back to equilibrium.

The well known radiative relaxation time, $\tau_R$ which is explained in a number of texts, is deduced for small perturbations in the temperature, i. e. it is assumed that the equilibrium has been slightly perturbed. Furthermore, $\tau_R$ depends on the values of p and T where it is computed. This is illustrated, for example, in Flasar et al. (1981) Table 3, where radiative times are calculated for several pressures. Other clear example is in Venus, where $\tau_R$ varies from 116 days at 40 Km (lower cloud deck) to 0.5 hr at 100 Km (Sanchez-Lavega et. al. (2017)).

On the contrary, in the computation of $\tau$, the only dependence is on globally averaged planetary parameters.

So, the difference between $\tau_R$ and $\tau$ is clear; $\tau_R$ is the relaxation time for small or moderate radiative perturbations, whilst $\tau$ is related to global perturbations.

It is important to keep in mind that the departures from equilibrium in the troposphere are damped by convection. At these pressures $\tau_R$ is very high, which makes radiative transfer inefficient. See for example Houghton's book, page 15.

Following your suggestion, a new section dealing the radiative timescale has been added.

We acknowledge your comments and dedication.

**6.1 Detailed comments:**

*P.2 Eq (7) - This assumes a simple integration with height, but atmospheres also vary in structure horizontally. Won't this make a difference?*

We have tested the accuracy of Eq (7) to reproduce the values of S shown in the literature (Peixoto and Oort (1992)). The agreement is excellent.

*Section 2 - By focusing on E or S as the main measures of energy you focus on essentially the dry static energy, which is dominated by internal energy. But much of this energy will be unchanged by internal dynamical adjustments and would be unlikely to vary unless the global thermal perturbation was to be fairly cataclysmic. Why is this the most significant quantity to calculate?*

We agree, if S is not much bigger than K+L, our result would be just a poor lower bound. Future observations could settle this result.

Let us remember K/S $\approx$ (wind speed / sound speed)$^2$. By using the highest wind speed recorded in Titan of 120 m/s at an altitude of 120 Km (see for example Bird, M., Allison, M., Asmar, S. et al. The vertical profile of winds on Titan. Nature 438, 800–802 (2005). https://doi.org/10.1038/nature04060), one obtains K/S $\approx$ 0.2.

Knowing that wind speeds in Titan decrease steadily as altitude decreases (see Bird el al. (2005)) we are confident that K/S is everywhere very small and the same can be said of Mars.

*Table 2 - It is mentioned that most of these figures for fluxes originate from the Trenberth diagrams published by Read et al. (2016). But the fluxes quoted appear to represent either the upward or downward IR fluxes between the atmosphere and surface. Would it not be more meaningful to compute the net flux entering or leaving the atmosphere? For Venus this would look more like 22 W/m$^2$ at the surface. The corresponding figure for Mars would be nearer 26 W/m$^2$ and 0.26 W/m$^2$ for Titan, based on the information in Read et al. (2016). These figures definitely need more explanation and justification.*

In Section 3, we have added a new clarifying paragraph showing how the fluxes $F_i$ and $F_o$ are calculated for Venus.

---

## Author Response (AR2)

In this version we have tried to meet both referees and editor criticisms and suggestions.

We have included uncertainties in energy fluxes and some bounds on latent and kinetic energies in the planetary atmospheres as requested by Rev. 2.

Section 4 has been changed and reduced.

---

## Author Response (AR3)

Dear Editor,
You can find attached two documents: a new version of the paper and a version where the modifications with respect to the old version are marked.

The introduced modifications try to meet the criticisms raised by the referees and what you pointed in your communication.
Specifically

> Editor's comments:
> * I feel Sections 4 and 5 may need some introductory and concluding remarks to make them better integrate with the rest of the study. Why these items are discussed and what the implications are for the study may need to be better outlined at the start and end of these sections.

Following the comments of referee #1, we have simplified Section 4. We have explicitly introduced the analogy between KH timescale and the residence time of energy in the planetary atmospheres. Also, we have modified Section 5 to justify the introduction of this Section.

> Rev. 2 also commented on the lack of a critical review of possible uncertainties in the input numbers (flux and energy content) you use in your calculations, and their effect on the results, this is something you may want to respond to.

Comparison between the different forms of energy and considerations about their relative magnitudes have been introduced in Table 2 (new) for Earth. Section 3 has been rewritten to include in the text quotes and figures about the uncertainty of energy fluxes. We also discuss some approximations that were made for Titan and Mars in lines 53-56. These uncertainties have been summarized in Table 4.

We look forward to hearing positive news from you

Sincerely

The authors.

---

## Author Response (AR4)

Dear Editor,

We have uploaded the manuscript and a track-changes file where the modifications with respect to the version uploaded on May 11[th] are marked.
We have included the changes from May's version in order to clarify many of the changes that we implemented to solve the issues you raised in your response of 28 June.

These changes are as follows:

> Equation 2: Although one can understand from the text that the F here is different from F in Eq. 1, but again, for consistency, use a different notation for each.

The font in Eq. 1 and the corresponding references in the text have been changed to emphasize the difference.

> Lines 21-23: first, I don't think there is a need for a new paragraph. Second, I understand why you keep the sentence, "But first it is worth recalling that several authors have previously considered the energy-residence time relation in other type of problems." However, in its current form, it feels unrelated and awkward. Either elaborate on this point or move it to the next paragraph to explain why you describe the residence timescale in the sun.

This paragraph has been modified.

> Equation 8: I do not see how this contributes to the manuscript.

Eq. 8 has been supressed.

> Line 67: no need to be in a separate paragraph.

Change accepted.

> Section 3 title: Why did you add Residence time at the end of the title? Maybe and residence time in planetary atmospheres?

Section 3 title has been changed to "Energy fluxes absorbed and emitted by the planetary atmospheres and residence times."

> Line 73: The case of Venus is Fig.6 in Read et al. 2016, not 3.

Yes, it was a mistake. Changed.

> Line 133: at the end of the line, change "The result" to "The radiative timescale …"

The paragraph has been re-written.

> I feel Sections 4 and 5 may need some introductory and concluding remarks to make them better integrate with the rest of the study. Why these items are discussed and what the implications are for the study may need to be better outlined at the start and end of these sections.

Following the comments of referee #1, we have modified Section 4. We have emphasized the analogy between KH timescale and the residence time of energy in the planetary atmospheres. Also, we have modified Section 5 to justify the introduction of this Section.

> Rev. 2 also commented on the lack of a critical review of possible uncertainties in the input numbers (flux and energy content) you use in your calculations, and their effect on the results, this is something you may want to respond to.

Comparison between the different forms of energy and considerations about their relative magnitudes have been introduced in Table 2 (new) for Earth. Section 3 has been rewritten to include in the text quotes and figures about the uncertainty of energy fluxes. We also discuss some approximations that were made for Titan and Mars in lines 53-56. These uncertainties have been summarized in Table 4.

---

## Author Response (AR5)

Dear editor:

We have uploaded the manuscript and a track-changes file from last version of the manuscript.

These changes are as follows:

Suggested change for the closing sentence of Section 1:

"In Section 6 we make some final comments."

Sections 5 and 6 have been modified, and the end of Section 1 has been modified accordingly.

Above Eq 3:

"these magnitudes"

Replace by " the first two quantities"

We agree. It has been modified.

Below Eq. 7, you say:

"the four terms U, P, K, and L are known (Peixoto and Oort, 1992), so we know E"

Suggest that you replace this with: "the four terms U, P, K, L, and hence E are well approximated (Peixoto and Oort, 1992)"

We agree. It has been changed.

l.61, "The so called 'Trenberth diagrams'" - "so-called"; also, can you please provide a reference?

We have included as reference the original paper from Trenberth, and also, the name "Trenberth diagram" appears in the title of the paper from Read et al. (2016) where the data from fluxes come from.

l.67, "160Wm−2, of longwave radiation" - delete comma

We agree. Clearly a misprint.

l.71: change "can be neglected" to "is small". For Venus, for example, the uncertainty is +/- 7%, not negligible.

We agree. Changed.

.73-84 - Can you please shorten this paragraph by paraphrasing key points from the Reid et al quote? The long quotation is unusual and not really necessary.

We agree. The quote has been reduced to a sentence.

The revised title for Section 3 is somewhat awkward. "Absorbed and emitted energy fluxes and residence time in planetary atmospheres" would read better, if it expresses what you intend to say?

We agree. The title of the Section has been changed.

Sections 5 and 6 - This comment is somewhat related to Rev. 1's criticism about Sections 4 and 5. While you did a good job connecting Section 4 to the rest of the study, I find Sections 5 and 6 lacking. It could be possibly valuable to relate your energy residence time to relaxation timescale. However, I find the outcome of the discussion, the final sentence of Section 5, not convincing. The link between energy residence time and relaxation time is not clarified or motivated well enough. There are three options: you drop references to relaxation time; mention relaxation time, but admit a clear link with energy residence time is not obvious at this time; or offer a more convincing explanation how the two are related. In any case, I would suggest you consider combining Sections 5 and 6, perhaps with a title "Discussion" or something similar? Section 6 as it is now does not constitute a section of its own. Incorporate l.145-149 into the part discussing relaxation time.

First sentence of Section 6 - this is a subjective statement that, without additional supportive argument or facts, is redundant. The readers can confidently assume that you believe energy residence time is a new concept, otherwise you would not publish about it. A point like this is fine in arguments with Referees but would be better avoided in a published paper.

Sections 5 and 6 have been merged and transformed into "Final discussion" in order to connect and compare energy residence time, $\tau$, with the often found radiative relaxation time, $\tau_R$.